Predicting student success in MOOCs: a comprehensive analysis using machine learning models

Althibyani Hosam A. halthibyani@uj.edu.sa
Learning Design and Technology Department, College of Education, University of Jeddah , Jeddah , Saudi Arabia
Lara Juan
Electronic publication date: 2024 Aug 23
Publication date: 2024
Volume: 10
Electronic Location ID: e2221
Received 2024 Mar 29; Accepted 2024 Jul 8
Copyright: ©2024 Althibyani
Copyright year: 2024
Copyright holder: Althibyani
License: This is an open access article distributed under the terms of the Creative Commons Attribution License, which permits unrestricted use, distribution, reproduction and adaptation in any medium and for any purpose provided that it is properly attributed. For attribution, the original author(s), title, publication source (PeerJ Computer Science) and either DOI or URL of the article must be cited.
License URL: https://creativecommons.org/licenses/by/4.0/

Keywords: Artificial intelligence, Logistic regression, Machine learning, OULAD, Random Forest, MOOC, Virtual learning environment

Funding: The authors received no funding for this work.

==============================
Background

This study was motivated by the increasing popularity of Massive Open Online Courses (MOOCs) and the challenges they face, such as high dropout and failure rates. The existing knowledge primarily focused on predicting student dropout, but this study aimed to go beyond that by predicting both student dropout and course results. By using machine learning models and analyzing various data sources, the study sought to improve our understanding of factors influencing student success in MOOCs.

Objectives

The primary aim of this research was to develop accurate predictions of students’ course outcomes in MOOCs, specifically whether they would pass or fail. Unlike previous studies, this study took into account demographic, assessment, and student interaction data to provide comprehensive predictions.

Methods

The study utilized demographic, assessment, and student interaction data to develop predictive models. Two machine learning methods, logistic regression, and random forest classification were employed to predict students’ course outcomes. The accuracy of the models was evaluated based on four-class classification (predicting four possible outcomes) and two-class classification (predicting pass or fail).

Results and Conclusions

The study found that simple indicators, such as a student’s activity level on a given day, could be as effective as more complex data combinations or personal information in predicting student success. The logistic regression model achieved an accuracy of 72.1% for four-class classification and 92.4% for 2-class classification, while the random forest classifier achieved an accuracy of 74.6% for four-class classification and 95.7% for two-class classification. These findings highlight the potential of machine learning models in predicting and understanding students’ course outcomes in MOOCs, offering valuable insights for improving student engagement and success in online learning environments.

Introduction

MOOCs have gained popularity over the years as a flexible and convenient mode of education (Virani, Saini & Sharma, 2023). These online courses offer learners the opportunity to access high-quality educational resources from top universities and institutions across the globe without having to attend a physical classroom. MOOCs also provide a cost-effective means of obtaining professional certifications or degrees (Yousef & Sumner, 2021). However, despite the numerous benefits of MOOCs, student dropouts remain a significant concern. Studies suggest that the dropout rates for MOOCs can range from 80% to as high as 95% (Mourdi et al., 2019). Moreover, students who are enrolled in technical educational programs often encounter difficulties in passing these courses and are at a higher risk of dropping out of their studies (Ouyang, Zheng & Jiao, 2022). Several studies have investigated student attrition or persistence and have primarily emphasized personal characteristics over academic performance, understanding students’ learning progress can benefit both learners and instructors (Bağrıacık Yılmaz & Karataş, 2022). By gaining insight into the learning process, students may be motivated to engage more thoroughly with the study material, while instructors can allocate resources more effectively to those who require additional support (Clarin & Baluyos, 2022). At the most basic level, Virtual Learning Environments (VLEs) e.g., MOOC platforms capture detailed data on video views, content pages accessed, quiz attempts and scores, forum activity, assignments completed and grades. Analytics dashboards parse this behavioral data to show student progress, participation, and achievement. Instructors can view overall class trends and statistics, while students can monitor their own progress versus benchmarks (Paiva, Leal & Figueira, 2022). Data mining techniques like clustering, classification trees and regression seek useful correlations between student behaviors and performance. This can build predictive models to identify at-risk students earlier. For instance, machine learning algorithms might predict students likely to drop out based on early participation patterns, allowing personalized retention interventions (Hasan et al., 2020).

The purpose of this study is to explore the potential of final exam outcomes as a valid predictor of student performance in online courses. By analyzing data from student profiles and interactive online activities within a learning ecosystem, the study aims to predict a student’s final grade. The utilization of the Open University Learning Analytics Dataset (OULAD), a comprehensive dataset from a prominent UK academic institution, allows for a robust analysis of student characteristics and self-assessed measures of learning progress (Kuzilek, Hlosta & Zdrahal, 2017). The study utilized a methodological approach involving the development of a multiple logistic regression model and a Random Forest Classifier to predict students’ final exam scores. The models serve as valuable tools for identifying students who may be facing challenges and are at risk of dropping out, enabling early intervention and support measures to be implemented.

Literature Review

Predicting student success and identifying factors that contribute to student performance in Massive Open Online Courses (MOOCs) has been an active area of research in recent years (Lemay & Doleck, 2022). One of the earliest studies in this domain was conducted by Alamri et al. (2021), who utilized logistic regression to predict student dropout in MOOCs. They found that features related to student engagement, such as the number of video lectures watched and the number of forum posts, were important predictors of student attrition. Building upon this work, further studies have explored more advanced machine learning algorithms. Benoit et al. (2024) applied Hidden Markov models to model student engagement patterns and predict their likelihood of dropping out. Their results showed that this approach outperformed logistic regression in terms of dropout prediction accuracy.

In a comparative analysis, Alsariera et al. (2022) evaluated the performance of several classification algorithms, including decision trees, random forests, and support vector machines, for predicting MOOC student performance. They found that random forests achieved the highest accuracy in predicting whether a student would pass or fail a course.

More recently, researchers have incorporated deep learning techniques into MOOC student success prediction. Won et al. (2023) developed a neural network model that leveraged both student activity data and text-based features from discussion forums to predict student dropout. Their model demonstrated superior performance compared to traditional machine learning approaches.

Additionally, some studies have explored the use of ensemble methods to combine the strengths of multiple algorithms. Niyogisubizo et al. (2022) proposed an ensemble of logistic regression, random forests, and gradient boosting models, which outperformed individual models in predicting student outcomes.

While these studies have contributed valuable insights, the literature review also reveals gaps in the existing research. Many of the previous works have focused on predicting student dropout, but there is a need for more comprehensive studies that also address the prediction of student performance in terms of final course grades or pass/fail outcomes. Furthermore, the majority of the existing literature has relied on a limited set of data sources, primarily focusing on student interaction and engagement metrics. There is an opportunity to explore the integration of additional data, such as daily activities information and assessment-related features, to provide a more holistic understanding of the factors influencing student success in MOOCs.

The current study aims to address these gaps by developing accurate predictive models that go beyond dropout prediction and incorporate a wider range of data sources to provide comprehensive insights into MOOC student performance. By leveraging both traditional and advanced machine learning techniques, this research seeks to advance the state of the art in MOOC student success prediction and contribute to the ongoing efforts to improve the quality and effectiveness of online learning environments.

Leading universities have shared anonymized data with researchers to facilitate important knowledge discovery. The Open University UK, with roughly 170,000 students enrolled in diverse programs, is the largest academic institution in the UK. Course materials are distributed via a VLE, and the OULAD dataset includes information from 32,593 students and 22 module-presentations, such as student demographic data, assessment dates and scores, and clickstream data detailing their usage of the VLE. This section presents different theories that exist in the literature regarding efforts that aimed to forecast a student’s course outcome, including passing or failing, or their likelihood of dropping out or successfully completing the course, are presented. The following is a brief description of some machine learning models utilized by the researchers and the relevant evaluation parameters they employed.

Haiyang et al. (2018) conducted a study utilizing the OULAD dataset, which consisted of records from 32,593 students and their interactions within the Virtual Learning Environment (VLE). The researchers focused on extracting daily clickstream data from the VLE and converting it into time series data frames for each module. The VLE interaction log contained a vast amount of information, with over 10 million entries. To forecast student dropout, the researchers employed Time Series Forest, deliberately excluding the candidate’s demographic information to address ethical concerns. In their investigation, Haiyang et al. (2018) classified the course materials provided by the Open University (OU), such as video lectures and readings, as Dropout and No Dropout data, achieving an accuracy of 93%. Additionally, they emphasized the importance of analyzing the data on submitted assignments, noting that students who failed to submit assignments had a 90% chance of dropping out from the course (Hlosta, Zdrahal & Zendulka, 2017) The prior educational background of students was also identified as a significant factor. Regarding the feature extraction process, Haiyang et al. (2018) utilized the OULAD dataset to derive features from the VLE environment. On the other hand, Hlosta, Zdrahal & Zendulka (2017) employed existing features such as the number of consecutive days a student was active, the average median clicks, and the number of materials visited per day.

Alshabandar et al. (2018) demonstrated that utilizing assignment due dates and the Virtual Learning Environment (VLE) database can effectively predict student dropout. The study employed a probabilistic model that combined various VLE data types for each student over a predetermined period. This included dynamic behavioral features, demographic variables, and various assignment features, which were combined into a single value. The prediction task used three algorithms in a Gaussian Finite Mixture model: Analysis via Discriminant Eigenvalue Decomposition (EDDA), K-nearest Neighbor (KNN), and logistic regression (LR). The study achieved an accuracy of 92%, 91.1%, and 89.3%, respectively, in dividing students into Dropout and No Dropout groups.

The XuetangX (OpenEdx) dataset, which is comprised of information on course content, student enrollment, and learning access logs, was also utilized by Hong, Wei & Yang (2017). This dataset was combined with other datasets to create a comprehensive analysis of student dropout behavior in massive open online courses (MOOCs). The study pooling these datasets found that as the assignment due date approached, more students withdrew from the class than submitted their assignments. The authors were able to accurately predict dropouts and non-dropouts with an accuracy rate of 86.1% and 86.5%, respectively, using support vector machines (SVM) and Random Forest (RF) methods.

In a related study, Xing & Du (2018) employed the Knowledge Discovery in Databases (KDD) Cup 2015 dataset, which consisted of 39 courses and seven CSV data files. The dataset was divided into training and testing sets, and the researchers found that the dropout rate for MOOC courses was significantly higher than that of traditional campus courses. The accuracy of the classification of students into dropout and no dropout groups by the researchers using k-nearest neighbor (KNN), SVM, decision tree (DT), and deep learning (DL) algorithms was 96.6%, 94.4%, 98.2%, and 97.4%, respectively. In previous work, researchers have employed Random Forest models to forecast student success using inputs such as lectures, quizzes, laboratories, and videos obtained from Moodle records. These models have been shown to achieve an accuracy rate of 96.3%. Recently, Ljubobratović & Matetić (2019) investigated the impact of laboratory and questionnaire results on final grades and found that these factors had the greatest influence on student performance. Similarly, Jha, Ghergulescu & Moldovan (2019) examined the effectiveness of various attributes, including demographic information, assessment results, and engagement with a virtual learning environment (VLE), in predicting both student dropout and academic performance. Their analysis of the OULAD dataset revealed that models based on students’ interactions with the VLE achieved the best performance, with AUC scores ranging from 0.91 to 0.93 for dropout prediction and result prediction using Gradient Boosting Machine.

In a similar vein, Balabied & Eid (2023) employed the random forest classifier model on extensive datasets from Open University Learning Analytics (OULAD). Their study achieved a 90% accuracy rate in identifying at-risk students, facilitating timely interventions that boost their academic performance. This approach significantly enhances the learning experience by providing early support and interventions tailored to students in need.

These results demonstrate the potential of machine learning techniques in predicting student dropout behavior in MOOCs. By analyzing large datasets containing various types of information about students’ interactions with the course material, these studies have shown that it is possible to identify patterns and trends that can help educators anticipate and prevent student dropout. However, it is important to note that these findings should be interpreted with caution, as the accuracy of the predictions may vary depending on the specific dataset and the machine learning algorithm used.

Research Method

To forecast the final grades of the OULAD, this investigation has developed two machine learning models: Logistic Regression and Random Forest.

Dataset preparation

Following a comprehensive analysis of the OULAD dataset, which consists of seven CSV files, it was determined that this dataset would be suitable for analysis. The dataset contains various types of information related to courses, students, registrations, assessments, online learning resources, and student interactions with the Virtual Learning Environment (VLE) as depicted in Fig. 1. The dataset encompasses essential information such as demographic data, assessments, and student interaction with online learning resources. Additionally, it includes assessment results submitted by students and detailed VLE clickstream data. These attributes can be leveraged to engineer features and construct models for predicting student performance in the course (Kuzilek, Hlosta & Zdrahal, 2017). Specifically, the dataset consists of the following files:

Figure 1 OULAD database schema.

1. The “courses” file, which contains 22 records and provides information about the courses.

2. The “studentInfo” file, comprising 32,593 records, contains demographic data about the students.

3. The “studentRegistration” file includes 32,593 records and provides information about the registration of each student for a course presentation.

4. The “assessments” file contains 196 records and provides information about assessments for each course presentation.

5. The “studentAssessments” file includes 173,740 records and provides information about the assessments submitted by the students.

6. The “vle” file consists of 6,365 records and contains information about online learning resources and materials.

7. The “studentVle” file includes 1,048,575 records and provides information about students’ interactions with the VLE resources.

Data analysis and feature extraction

At first, beneficial attributes were identified, including the score in the studentAssessment table and the sum-click in the studentVle table. As a result, the sum-click and score features were grouped, and the total portal clicks for a particular student throughout the year and their average grade were calculated. The researcher predicted that these features would demonstrate a positive correlation since better scores and grades are generally linked to increased effort. Afterward, I graphed the information and observed that a logistic regression model would be highly effective. Then incorporated additional data into the model, aiming to utilize as much data as possible to assist the model in identifying patterns. For instance, computed the number of days a student submitted their coursework early and their summative and formative marks (weight = 0). Since more prepared and committed students are likely to perform better, as anticipated a positive correlation from this data. As my understanding of the data progressed as I integrated a broad set of available data, causing me to interpret the information in a different way. This involved computing the mean, median, mean absolute deviation, standard deviation, and variance for various data in the schema. Furthermore, correlation heatmap was created as depicted in Fig. 2 and sorted numerical correlations in Table 1 to gain further insight into the data.

Figure 2 Correlations heatmap.

Table 1 Numerical correlations.

Feature	Correlation	
daysEarlystdScore	−0.259014	
studied-credits	−0.176016	
region-Wales	0.008382	
age-band	0.068551	
score	0.317339	
sum-click	0.376107	
totalCoursework	0.427175	
summativeAgainstCredits	0.490646	

It is unexpected that the age-band reveals a weak correlation, as one would typically expect a slight negative correlation. However, this might be due to the limited amount of data available, with just three distinct ranges depicted in Fig. 3. To enhance this correlation, it would be necessary to expand the ranges or employ integers instead of ranges. Following the data collection phase, we processed the data using an imputer and scalar. The imputer replaced all missing values (NA) with the median value of that specific feature. The scalar normalized the features within the range of 0 to 1, preventing large ranges from dominating the features and ensuring that the features were unit dependent. Furthermore, the transformation of region, code module, and code presentation into columnar data was performed using one-hot encoding for these categories.

Figure 3 Linear regression (two classes) against age-band.

Model selection

The model selection stage was focused on, starting with the division of the data into train and test sets using a 75/25 split. Subsequently, numerous models were tested, and their performance was compared through cross-validation on the training data. A wide range of performance was observed among the models, with classifiers generally exhibiting better results than regressors. For further details, Table 2 should be consulted. It was decided to investigate one regressor and one classifier for further analysis: logistic regression and Random Forest Classifier.

Table 2 Model selection cross-validation accuracy metrics.

Model	Classes	Mean	SD	
Linear regression	4	0.604	0.010	
Logistic regression	4	0.702	0.007	
Logistic regression	3	0.767	0.005	
Logistic regression	2	0.915	0.003	
SVR linear	4	0.575	0.013	
SVR poly	4	0.746	0.008	
SVR RBF	4	0.723	0.009	
SVC	4	0.681	0.007	
SVC	3	0.782	0.005	
SVC	2	0.932	0.003	
DT	4	0.674	0.002	
DT	3	0.757	0.006	
DT	2	0.916	0.004	
RF	4	0.744	0.006	
RF	3	0.813	0.004	
RF	2	0.942	0.003	
Notes.

SVR, Support Vector Regression; SVC, Support Vector Classifier; DT, Decision Tree Classifier; RF, Random Forest Classifier.

Logistic regression model

The initial part of this section introduces fundamental concepts in logistic regression, such as the polytomous model, logit model, probit model, tobit model, and logistic function. The section then covers various issues related to analysis and interpretation, for which established standards are presented. These include topics such as the minimum observation-to-predictor ratio, assessment of potential interaction effects, evaluation of logistic regression models, diagnostic statistics, and three reporting formats for logistic regression outcomes. A typical regression model generally has the following appearance:

y ˆ = b0 + b1x1 + b2x2 + ………+ bpxp

The predicted value of the outcome variable, denoted by y-hat, is estimated based on the true value of Y, where b0 is the constant term in the equation and b1, b2, …, bp represent the estimated parameters associated with predictor variables x1, x2, …, xp . The Y-intercept is another term for b0, while b1, b2, …, bp can also be referred to as slopes, regression coefficients, or regression weights (Peng et al., 2002).

The least squares method is commonly employed by statisticians to estimate parameters and obtain least squares estimates. However, when dealing with categorical outcome variables like whether a high school graduate matriculated into college, this method proves inadequate. In such cases, the data plot exhibits parallel lines, each corresponding to a specific value of the outcome variable (e.g., 1 = matriculated, 0 = did not matriculate). Consequently, the variance of residuals becomes a function of the proportion of a particular outcome at specific X values. The categorical nature of the outcome variable renders it impossible to fulfill the normality assumption for residuals or the assumptions of continuity and unboundedness for Y. As a result, significance tests on regression coefficients are invalid, although the least squares estimates retain their unbiased nature. Even if categorical outcomes are redefined as probabilities, predicted probabilities from least squares regression models can sometimes exceed the logical range of 0 to 1.

Hence, alternative methods such as logistic or probit regression are necessary for modeling categorical outcome variables. The issue arises because the model lacks a mechanism to constrain predicted values, which may lead to illogical values outside the expected range. Additionally, the R2 index, typically used to indicate the amount of variance explained in a model, does not carry the same meaning for categorical outcomes when employing least squares regression. It cannot be utilized to assess predictive efficiency or be tested within an inferential framework (Menard, 2000).

Various alternative statistical techniques have been proposed to overcome the limitations of least squares regression when handling categorical variables. These techniques include discriminant function analysis, log-linear models, linear probability models, and logistic regression. Logistic regression is considered superior to other techniques as it can accommodate both continuous and discrete predictors, is not constrained by normality or equal variance/covariance assumptions for the residuals and is connected to discriminant function analysis through the Bayes theorem. Logistic regression has also demonstrated fairly accurate results in classification and prediction tasks (Flury, 2013).

The logistic regression model estimates the probability (P) of the outcome of interest, also known as the ”event”, given the variable Y. The Y intercept (α) and the slope parameter (β) are both estimated using the maximum likelihood (ML) method, which aims to maximize the likelihood of obtaining the observed data given the parameter estimates. The relationship between the parameters (α, β) and the probability of observing a particular outcome, such as a high school graduate matriculating, in an observation is nonlinear. However, the relationship between (α, β) and the logit is linear. When there is only one predictor X, the logistic regression curve takes the form of a normal ogive, as shown in Eq. (1).

(1) LnP/1−P=logodds=logit=α+βx.

In logistic regression, the inferential framework revolves around the null hypothesis, which assumes that the population parameter β equals zero. Rejecting this null hypothesis indicates a relationship between the predictor variable X and the outcome variable Y. When the predictor variable is binary, such as gender, the exponentiated β (e ˆ β) represents the odds ratio, which signifies the ratio of the two odds. For instance, if the odds for girls to matriculate are 1.5 times more (or less) likely than the odds for boys, the odds ratio can be utilized to interpret the relationship between gender and matriculation.

When the predictor variable is continuous, Chen, Song & Ma (2019) suggests employing delta-P (change in probability) to interpret the results of logistic regression. These two interpretations are frequently utilized by authors in published articles and are relatively straightforward, as evidenced by the examination of 52 articles. In this study, logistic regression was employed to classify data into two scenarios. The first scenario involved classifying the data into two categories: “fail” and “pass”. The second scenario involved classifying the data into four categories: “fail”, “pass”, “withdrawn”, and “distinction”. These classifications were based on data analysis, feature extraction, and are illustrated in Fig. 4.

Figure 4 Logistic regression and random forest architecture.

Regarding hyper parameter tuning, we opted to utilize a grid search to identify the optimal combination within the given domain for this model. We examined a logarithmic range of the C parameter and found the optimal range to be between 950 and 1,100 after conducting tests. Additionally, we evaluated tolerance values around the default and determined that 0.0015 was the most effective. Another set of combinations was tested, which involved examining the same values of C while adjusting the solver and penalty used. However, we later discovered that this second set of combinations did not improve performance and thus eliminated it from consideration.

Random Forest Classifier model

The Random Forest Classifier algorithm builds multiple decision trees, each trained on different subsets of the same training data, to improve classification accuracy and mitigate the risk of overfitting (Parmar, Katariya & Patel, 2019). By using an ensemble of decision trees, the classifier can reduce variance and improve its ability to generalize to new, unseen data. The Random Forest algorithm randomly selects a subset of attributes to create K decision trees without pruning. In contrast, a decision tree is constructed from the training data and used to predict the outcome of the test data. However, in Random Forest, the test data is passed through all the decision trees, and the most frequent output is assigned to the instance (Mishra, Kumar & Gupta, 2014). This approach is designed to improve the accuracy and stability of the classifier by reducing the impact of individual decision trees and incorporating their collective decision-making power.

In general, a Random Forest Classifier is more robust when it has a greater number of trees in the forest (Balabied & Eid, 2023). This technique offers high accuracy when more trees are included in the forest. Additionally, Random Forest can handle missing values and categorical variables, while avoiding overfitting. To measure the purity and impurity of attributes, Random Forest utilizes the Gini index indicator (Charles, Gherman & Paliza, 2022).

Gini Index

The Gini Index is an attribute selection method that evaluates the attribute purity or homogeneity with respect to class values in a dataset. Like entropy, the Gini Index reaches its highest value when the probabilities of each class are equal. The Gini Index quantifies attribute impurity using the following formula (Pal, 2005). Ginit=1−−∑i=0c−1pi|t2

In this investigation, Random Forest was used to categorize data according to two different scenarios. The first scenario aimed to divide the data into two classes: “pass” and “fail”. In the second scenario, the data was categorized into four classes: “pass”, “fail”, “withdrawn”, and “distinction”. The classification was determined by analyzing the data and extracting features from data as shown in Fig. 4.

At first, we employed Random Search to adjust the parameters of the classifier, which produced parameters for model cross-validation. The study focused on exploring several parameters in the model, including the number of estimators, the maximum depth, the minimum number of samples required for a leaf node, and the minimum number of samples necessary to split an internal node. We subsequently switched to using Bayesian optimization to optimize the search problem. This approach utilizes previous iterations to strategically select the best parameters from the search space to minimize the loss function, which we defined as 3−acc¯−accbal¯−f¯1weighted. This function aims to minimize the primary metrics for a classification problem. Additionally, we eliminated unimportant attributes from the forest’s feature importance’s and returned the model.

Results

In this study the OULAD dataset was utilized, which contains information on 32,593 students and includes various attributes (see Table 1). The objective was to determine the best classification model for predicting student performance on the final exam, so they utilized two models: logistic regression and Random Forest. All attributes in the dataset are independent, except for the class attribute (output), which depends on all of them. Therefore, the dataset was trained on both models using the Python language in the Anaconda Navigator program. The researcher then compared the results of the two models to determine which one was more accurate.

Logistic regression model results

The logistic regression model has a lower weighted average of the F1 score, and this metric is likely better for measuring performance as it can account for data imbalance. The analysis of the classification report suggests that classes with lower balance have worse performance due to the data imbalance. The confusion matrices indicate that similar classes such as withdrawn and fail and pass and distinction are often confused. Additionally, the two-class metrics, which combine withdrawn and fail, and pass and distinction, typically have strong performance as seen in Tables 3 and 4.

Table 3 Logistic regression four classes confusion matrix.

	Withdrawn	Fail	Pass	Distinction	
Withdrawn	2,096	374	72	2	
Fail	724	641	333	3	
Pass	84	125	2,748	164	
Distinction	1	3	390	389	

Table 4 Logistic regression two classes confusion matrix.

	Fail	Pass	
Fail	3,889	356	
Pass	261	3,643	

Random Forest classifier results

After applying the Random Forest model to the data to classify it into two categories, the first category including classes such as “withdrawn”, ”fail”, “pass”, and ”distinction”, and the second category including classes “fail” and “pass”. The confusion matrices (Tables 5 and 6) revealed that similar classes such as “withdrawn” and “fail” and “pass” and “distinction” were often confused with each other. However, the metrics for the two-class model (which combines “withdrawn” and “fail” and “pass” and “distinction” as subclasses of the primary labels) were generally high-performing. Therefore, the Random Forest two-class model would be suitable for predicting whether a student is likely to fail, given its high accuracy.

Table 5 Random forest four classes confusion matrix.

	Withdrawn	Fail	Pass	Distinction	
Withdrawn	2,079	439	22	4	
Fail	620	803	275	3	
Pass	6	154	2,635	326	
Distinction	0	0	218	565	

Table 6 Random forest two classes confusion matrix.

	Fail	Pass	
Fail	3,952	293	
Pass	54	3,850	

Resolution

At the end of the evaluation, the logistic regression model achieved an accuracy of 72.1% for the four-classes classification task and 92.4% for the two-classes classification task. On the other hand, the random forest classifier achieved an accuracy of 74.6% for the 4-classes classification task and 95.7% for the 2-classes classification task. A comparison of the results for both models can be found in Table 7.

Table 7 The final metrics for the Random Forest and logistic regression models.

Model	Logistic regression	Random forest	
Classes	2	4	2	4	
Explained variance	0.697	0.664	0.833	0.734	
RMSE	0.275	0.585	0.206	0.519	
MAE	0.076	0.300	0.043	0.258	
r2 Score	0.697	0.662	0.829	0.734	
f1 Score (weighted) (recall and precision	0.924	0.706	0.957	0.742	
Accuracy	92.4%	72.1%	95.7%	74.6%	
Notes.

MAE, Mean Absolute Error; RMSE, Root Mean Square Error.

Discussion

The article introduced two supervised models, namely logistic regression and Random Forest, which employed classification techniques to forecast two outcomes: whether students would pass or fail their final exams, and whether they were susceptible to dropping out. Using the OULAD dataset, we developed classification models to forecast the academic performance of students. Our findings indicate that whether a student was active on a particular day has comparable predictive capability to a blend of detailed information, such as the exact number of clicks on that day, and private information, including gender, disability, and highest education level. Furthermore, the results presented in Fig. 5 provide valuable insights into students’ daily activity patterns and study preferences. The data shows that students tend to prefer studying during the morning hours rather than the evening, and they seem to favor shorter study sessions over longer ones. This helps explain the observed increase in task completion rates during the morning time.

Figure 5 Students’ daily activity patterns and study preferences.

These findings have important implications for designing more effective student support interventions. By focusing on students’ daily activity levels and preferred study times, educators and academic support services can tailor their approaches to better align with students’ natural behavioral patterns and preferences. This could lead to improved engagement, productivity, and ultimately, better academic outcomes.

According to the findings, the Random Forest model is the most effective in projecting which students are prone to dropping out from the course. This not only enables the identification of at-risk students with a high level of certainty but also allows for the anticipation of their performance in the final exams. This finding is consistent with that of Jha, Ghergulescu & Moldovan (2019) and Alshabandar et al. (2018) who confirmed that the Random Forest (RF) model achieved a low root mean square error (RMSE) of 8.131 for the students’ assessment grades model. On the other hand Gradient Boosting Machine (GBM) model yielded the highest accuracy of 0.086 for predicting final students’ performance.

The confusion matrices show that there is often confusion between comparable categories, such as withdrawn and fail, and pass and distinction. Moreover, the two-class metrics, which amalgamate withdrawn and fail, as well as pass and distinction, generally exhibit robust performance. These results corroborate the findings of a great deal of the previous proposed model utilizes feature selection methods and ensemble machine learning algorithms. It was tested on a dataset consisting of more than 5,500 learners from two MOOC courses at Stanford University. The model’s performance was evaluated, achieving a high accuracy rate of 98.6%. Additionally, a comparison was conducted using multiple performance measures to validate the model’s effectiveness (Mourdi et al., 2019). Thus, the study’s findings suggest that the developed predictive model can assist instructors in identifying learners at risk of dropping out, those likely to fail, and those progressing towards success in MOOCs. This classification enables targeted interventions and support strategies to be implemented, which may help mitigate the high dropout rates observed in MOOCs.

This study can aid high schools and universities in enhancing their performance by providing insights into students’ characteristics and identifying those who are at risk of dropping out. Additionally, it can anticipate students’ outcomes in their final exams, which can assist in supporting those who may be struggling. Furthermore, parents can benefit from this information to monitor their students’ academic progress. Future studies could explore the use of machine learning techniques for predicting student dropouts and academic performance in different contexts and with diverse datasets. These findings suggest that students’ prior assessment performance and engagement have a significant impact on their subsequent achievements in MOOCs. The models developed in this study can effectively detect the factors that influence student learning outcomes. By identifying these factors, educational institutions can implement targeted interventions and support mechanisms to improve student success rates in MOOCs. Additionally, the study did not investigate the ethical implications of using machine learning for predicting student outcomes, which is an important consideration in the field of education. Therefore, further exploration of the study would be necessary to gain a deeper understanding of the research methodology and the specific factors influencing student achievement in MOOCs. Moreover, researchers have recognized the need to address the class imbalance issue inherent in MOOC data, where the proportion of successful students is typically much higher than that of unsuccessful students. Shahabadi et al. (2021) proposed a novel ensemble method that combined Oversampling and Undersampling techniques to address this challenge, leading to improved predictive accuracy.

One key advantage of the models is their reliance on relatively simple and easily observable student data, such as daily activity levels and patterns of engagement. This type of information is often available in both online and in-person educational environments, making the models potentially transferable to different settings. For example, in a traditional classroom setting, instructors could track student attendance, participation, and the completion of assignments and in-class activities. This data could then be used to train predictive models similar to those developed in the study, to forecast student outcomes like pass/fail rates and risk of dropout. Additionally, the insights into students’ preferred study times and session durations could be applicable to both online and in-person learning contexts. Educators in traditional classrooms could use this information to optimize the timing and structure of their lessons, assignments, and support services to better align with students’ natural behavioral patterns and preferences.

However, it is important to note that the specific features and performance of the predictive models may need to be adapted and validated for different educational contexts. Factors such as the availability and reliability of data, as well as the unique characteristics of the student population, could impact the models’ effectiveness. Furthermore, ethical considerations, such as the potential for bias and the responsible use of student data, would need to be carefully addressed when applying these predictive models in any educational setting, whether online or in-person.

Conclusion

While MOOCs have gained popularity in recent years, high dropout and failure rates remain a challenge. Prior studies mainly focused on predicting dropouts. This study aimed to predict both student dropout and course results using logistic regression and random forest models built on various attribute categories including demographics, assessment scores, and student interaction data from the virtual learning environment (VLE). The study leveraged the OULAD dataset, which includes information not extensively examined previously. Logistic regression utilizing student VLE interaction attributes achieved 72.1% accuracy for four-class result classification and 92.4% for two-class. Meanwhile, random forest classification using these features reached 74.6% accuracy for four outcomes and 95.7% for two outcomes. Future work will concentrate on feature selection and engineering to further boost dropout and result prediction performance. A focus on incorporating time-sensitive metrics related to assessments and student engagement within the VLE may provide additional insights. Overall, this research demonstrates the potential of machine learning approaches to gain a more holistic view of MOOC student success when utilizing diverse interaction data.

Supplemental Information

Supplemental Information 1 Code using Python language

Additional Information and Declarations

Competing Interests

Author Contributions

Data Availability

The authors declare there are no competing interests.

Hosam A. Althibyani conceived and designed the experiments, performed the experiments, analyzed the data, performed the computation work, prepared figures and/or tables, authored or reviewed drafts of the article, and approved the final draft.

The following information was supplied regarding data availability:

The data is available at the anonymized Open University Learning Analytics Dataset (OULAD). The dataset consists of tables connected using unique identifiers. All tables are stored in the csv format: https://analyse.kmi.open.ac.uk/open_dataset.

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
