# Peer review of "Predicting student success in MOOCs: a comprehensive analysis using machine learning models"

_PeerJ Computer Science, doi:10.7717/peerj-cs.2221_

## Round 0.1 · original submission · Major Revisions

Please, address carefully all the reviewers' comments that have suggested Major Revision of your work. In particular, clarify the gaps that justify this work, complete the literature review and complete your discussion by comparing with existing works.

·

Basic reporting

no comment

Experimental design

Enhancement of Related Work
To improve the contribution related to the literature, a more comprehensive literature review that includes a wider range of algorithms previously used and compares their results with this study's findings would be beneficial. Additionally, identifying gaps in previous research and explaining how this study addresses these gaps will provide strong justification for its contributions.

Validity of the findings

Validation and Documentation

In line #55, it is stated that simple indicators such as a student's activity level on a particular day are as effective as more complex feature combinations. However, this claim lacks evidence in the document. To support this claim, the following is necessary:
1. Comparative Analysis: Include an analysis comparing the prediction results using simple indicators with those using more complex feature combinations.
2. Data Visualization: Provide data visualizations showing model performance using different feature sets.


In line #51, the study claims to go beyond merely predicting student dropout to predicting students' course outcomes (pass or fail). However, it does not explain what "going beyond" entails. To clarify, the study should detail how it not only predicts students' final outcomes but also offers insights into the factors contributing to their success or failure and how these findings can be used for more effective educational interventions.

Additional comments

Simplification and Combination of Figures
Regarding the figures, since Figure 4 (LR) and Figure 5 (RF) use similar flows, they can be combined into a single figure for efficiency. For example, a comparative figure showing the performance of Logistic Regression and Random Forest in different subplots within a single figure would provide a clearer and more concise representation.

·

Basic reporting

This paper predicts students’ MOOC outcomes using logistic regression and random forest classification by considering factors such as demographics, assessment, and student interactions.

The “Lay Summary” section could be edited into an abstract, which contains a brief introduction to the prediction tasks of MOOC outcomes and highlights the novelty of this research project: what types of experiments and methods are used, the conclusion as well as ablation studies(if any). The current section only touches on this topic briefly and does not make it suitable for scientific publication. From Line 51 to Line 57, the size of data sources and how they are obtained, as well as the interpretability of the third point could be expanded further.
From line 165: “By analyzing large datasets containing various types of information about students' interactions with the course material, these studies have shown that it is possible to identify patterns and trends that can help educators anticipate and prevent student dropout.”, and line 55 in Lay Summary section, “The findings of this study suggest that simple indicators, such as a student's activity level on a given day, can be as effective as more complex data combinations or personal information in predicting student success in MOOCs.”; It seems that factors such as student demographics do not play any important roles in predicting the course outcomes. It would be helpful to include a more comprehensive numerical correlation table in addition to Table 1 in the appendix.
Minor issues: From line 102, “and 22 module-presentations,”, there is no need to add hyphens.
Thanks for providing a comprehensive list of references.

Experimental design

In the “Data Preparation” section, it would be helpful to include an exploratory analysis to gain more in-depth insights into different variables. As the article later suggests, it is easy for prediction models to mix factors such as “fail” and “withdrawal”. Understanding relationships between each variable can help us to identify patterns and prepare for prediction tasks.
In the “Discussion” section, it is mentioned that “This study can aid high schools and universities in enhancing their performance by providing insights into students' characteristics and identifying those who are at risk of dropping out.”. I think the study needs more in-depth Interpretability analysis to make such claims as its performance and accuracy highly depend on the dataset. Such results are still very specific to MOOCs instead of full-time/part-time education institutions.

Validity of the findings

The findings of the paper are valid as it is supported by experiments conducted on open datasets. For a more accurate representation of the performance of the proposed model, please address comments in the Experimental Design section. The paper could also benefit from a more comprehensive literature review.

In the “Results and Conclusions” section, it would be helpful to address how much data is used for prediction tasks and if there are any potential biases.

Additional comments

Thanks for the opportunity to review the manuscript. Please address the comment above.

·

Basic reporting

Basic reporting is clear.

Experimental design

Research Question is well defined. The methodology section is detailed and well-structured describing the data preprocessing steps, feature selection, and model evaluation techniques

Validity of the findings

Raw data is provided. The results section is thorough.

Additional comments

Predicting student success in MOOCs: A Comprehensive analysis using machine learning models (#98470) Peer Review:

This article has many good points. The literature review is thorough. Research objective is very clear. The methodology section is detailed and well-structured describing the data preprocessing steps, feature selection, and model evaluation techniques. The findings of the study have significant practical implications for improving student engagement and success in MOOCs. The results section is thorough, presenting the performance metrics of both machine learning models in a clear and concise manner. Thank you so much for this wonderful and well researched article.


This article has below peer review comments upon review of this article.


1. The manuscript is clear. The methodology section could contain more information about Machine Learning for non-experts.
2. The research question is well-defined and relevant. Are there any existing gaps that could be explicitly stated which this study aims to fill?
3. Your introduction to background was great. You may need to include more recent studies on MOOC dropout rates and success predictors to provide a more current backdrop for the research.
4. The literature review is comprehensible. You may need to discuss different machine learning models used in similar studies. This would provide a broader perspective and justify the choice of logistic regression and random forest models.
5. In literature review you may need to discuss how the current study differentiates itself from previous research
6. OULAD Database schema - could be simplified for better clarity, and additional descriptive captions would help readers quickly grasp the key points.
7. Can you add more details to the metadata description for the datasets to enhance reproducibility and usability by others?
8. You may need to add why other potential machine learning models were not selected for the study.
9. Can you discuss potential impact of multicollinearity among predictors on the logistic regression model's performance?
10. The comparative analysis between logistic regression and random forest models would be insightful. That may include a discussion on the computational efficiency of each model would be beneficial for practical applications.
11. The study successfully demonstrates the applicability of the models on the OULAD dataset. Are there any potential limitations and challenges of generalizing these findings to other MOOCs?
12. The authors mention future research but could provide more concrete suggestions on specific areas where further investigation is needed, especially in improving model accuracy and handling different types of data.

---

## Round 0.2 · accepted · Accept

The reviewers have indicated that all their issues have been addressed properly so I recommend this paper for publication. If you can avoid personal ways such as "I" or "my" in the final manuscript it would be fantastic.

·

Basic reporting

No comment

Experimental design

The author has improved the related work section in the revised manuscript. The literature review is now more comprehensive, including a broader range of algorithms previously used in similar studies. Additionally, the author has meticulously compared these algorithms' results with the current study's findings. The revised section identifies gaps in the existing research and effectively explains how the current study addresses them, providing a strong justification for its contributions

Validity of the findings

The author has addressed the enhancement requirements for the related work section and included a comparative analysis in response to the request for evidence to support the claim that simple indicators, such as a student's activity level on a particular day, are as effective as complex feature combinations. This detailed comparative analysis compares the prediction results using simple indicators with those using more complex feature combinations. The author has provided this analysis in Figure 5.

Additional comments

In response to the suggestion to combine Figure 4 (Logistic Regression) and Figure 5 (Random Forest) into a single figure, the author has created a combined comparative figure. The new figure provides a clearer and more concise representation of the model performances by displaying the Logistic Regression and Random Forest results in different subplots within a single figure.

·

Basic reporting

The authors added numerical correlations of different features of the dataset.

Experimental design

The confusion matrices of classification categories such as withdrawn and fail are provided.

From line 462 to line 470, the paper discussed the limitations of the findings and how it does not generalize well to studies of other domains. The class imbalance of the MOOC data heavily influenced the prediction result. However, it does provide additional information for academic institutions to understand the student body.

Validity of the findings

Based on the study and evidence provided the findings can be considered valid as it is specific to a limited domain.

·

Basic reporting

This version looks good now. Thank you for fixing the comments.

Experimental design

This version looks good now. Thank you for fixing the comments.

Validity of the findings

This version looks good now. Thank you for fixing the comments.

Additional comments

This version looks good now. Thank you for fixing the comments.